# Pre-Diagnosis Diet Predicts Response to Exclusive Enteral Nutrition and Correlates with Microbiome in Pediatric Crohn Disease

**DOI:** 10.3390/nu16071033

**Published:** 2024-04-02

**Authors:** Stephanie Dijk, Megan Jarman, Zhengxiao Zhang, Morgan Lawley, Muzammil Ahmad, Ricardo Suarez, Laura Rossi, Min Chen, Jessica Wu, Matthew W. Carroll, Anthony Otley, Mary Sherlock, David R. Mack, Kevan Jacobson, Jennifer C. deBruyn, Wael El-Matary, Colette Deslandres, Mohsin Rashid, Peter C. Church, Thomas D. Walters, Hien Q. Huynh, Michael G. Surette, Anne M. Griffiths, Eytan Wine

**Affiliations:** 1Department of Physiology, University of Alberta, Edmonton, AB T6G 1C9, Canada; 2Department of Agriculture, Life, & Environmental Science, University of Alberta, Edmonton, AB T6G 2R3, Canada; m.jarman@aston.ac.uk; 3Department of Medicine, University of Alberta, Edmonton, AB T6G 2R3, Canada; zxzhang@jmu.edu.cn; 4College of Food and Biological Engineering, Jimei University, Xiamen 361000, China; 5Division of Pediatric Gastroenterology, Department of Pediatrics, University of Alberta, Edmonton, AB T6G 2R3, Canadamahmad1@ualberta.ca (M.A.); suarez@ualberta.ca (R.S.); mcarroll@ualberta.ca (M.W.C.); hien.huynh@ualberta.ca (H.Q.H.); 6Farncombe Family Digestive Health Research Institute, McMaster University, Hamilton, ON L8S 4L8, Canada; rossil@mcmaster.ca (L.R.); surette@mcmaster.ca (M.G.S.); 7Nutrition Services (Child Health), Alberta Health Services, Edmonton, AB T5J 3E4, Canada; min.chen@albertahealthservices.ca (M.C.); jessica.wu@albertahealthservices.ca (J.W.); 8Division of Gastroenterology & Nutrition, Department of Pediatrics, Dalhousie University, Halifax, NS B3H 4R2, Canada; arotley@dal.ca (A.O.); mohsin.rashid@iwk.nshealth.ca (M.R.); 9Division of Gastroenterology and Nutrition, Department of Pediatrics, McMaster University, Hamilton, ON L8S 4L8, Canada; sherlom@mcmaster.ca; 10CHEO IBD Center, Department of Pediatrics, University of Ottawa, Ottawa, ON K1N 6N5, Canada; 11Division of Gastroenterology, Hepatology and Nutrition, B.C. Children’s Hospital, British Columbia Children’s Hospital Research Institute, Vancouver, BC V5Z 4H4, Canada; kjacobson@cw.bc.ca; 12Section of Pediatric Gastroenterology, Department of Pediatrics, University of Calgary, Calgary, AB T2N 1N4, Canada; jennifer.debruyn@albertahealthservices.ca; 13Section of Gastroenterology, Hepatology and Nutrition, Department of Pediatrics, Max Rady College of Medicine, University of Manitoba, Winnipeg, MB R3T 2N2, Canada; welmatary@hsc.mb.ca; 14Division of Gastroenterology, Hepatology and Nutrition, Department of Pediatrics, CHU Sainte-Justine Hospital, Université de Montréal, Montréal, QC H3T 1J4, Canada; colette.deslandres@umontreal.ca; 15Division of Pediatric Gastroenterology, IBD Center, Hospital for Sick Children, University of Toronto, Toronto, ON M5S 1C6, Canada; peter.church@sickkids.ca (P.C.C.); thomas.walters@sickkids.ca (T.D.W.); anne.griffiths@sickkids.ca (A.M.G.); 16Michael G. DeGroote Institute for Infectious Disease Research, McMaster University, Hamilton, ON L8S 4L8, Canada

**Keywords:** pediatrics, inflammatory bowel diseases, nutrition, microbiome, dietary pattern, prediction

## Abstract

Exclusive enteral nutrition (EEN) is effective in inducing remission in pediatric Crohn disease (CD). EEN alters the intestinal microbiome, but precise mechanisms are unknown. We hypothesized that pre-diagnosis diet establishes a baseline gut microbiome, which then mediates response to EEN. We analyzed prospectively recorded food frequency questionnaires (FFQs) for pre-diagnosis dietary patterns. Fecal microbiota were sequenced (16SrRNA) at baseline and through an 18-month follow-up period. Dietary patterns, Mediterranean diet adherence, and stool microbiota were associated with EEN treatment outcomes, disease flare, need for anti-tumor necrosis factor (TNF)-α therapy, and long-term clinical outcomes. Ninety-eight patients were included. Baseline disease severity and microbiota were associated with diet. Four dietary patterns were identified by FFQs; a “mature diet” high in fruits, vegetables, and fish was linked to increased baseline microbial diversity, which was associated with fewer disease flares (*p* < 0.05) and a trend towards a delayed need for anti-TNF therapy (*p* = 0.086). Baseline stool microbial taxa were increased (*Blautia* and *Faecalibacterium*) or decreased (*Ruminococcus gnavus* group) with the mature diet compared to other diets. Surprisingly, a “pre-packaged” dietary pattern (rich in processed foods) was associated with delayed flares in males (*p* < 0.05). Long-term pre-diagnosis diet was associated with outcomes of EEN therapy in pediatric CD; diet–microbiota and microbiota–outcome associations may mediate this relationship.

## 1. Introduction

The worldwide incidence of inflammatory bowel diseases (IBDs) is increasing, especially in children [1]. Pediatric IBD can result in linear growth failure, delayed puberty, and reduced peak bone density [2]. The poorly understood etiology includes altered immune system, environment, gene, and gut microbiome interactions [3,4]. Diet, an important component of the gut environment, is one of the main determinants of the intestinal microbiome [5], which is significantly altered in IBD [6]. Diet is known to affect the risk of IBD development or flare [7,8] and is thought to exert effects through altering host immunity, intestinal barrier integrity, and microbiota [7].

A Mediterranean diet (MED) high in vegetables, fruits, cereals, and olive oil can reduce disease activity in both forms of IBD—Crohn disease (CD) and ulcerative colitis (UC) [9,10]. The MED increases microbial taxa associated with health, such as *Faecalibacterium prausnitzii* and *Roseburia* spp. [11]; both of these are reduced in IBD [6]. Mediterranean and other dietary-fiber-rich diets can reduce the risk of host intestinal mucus layer depletion by microbes, compromising intestinal barrier integrity and increasing immune activation [7].

Exclusive enteral nutrition (EEN) entails 6–8 weeks of liquid meal replacement with the exclusion of all other food and is a first-line therapy for mild to moderate CD [12]. EEN is more effective than glucocorticoids (GCSs) for achieving mucosal healing, inducing remission in up to 80% of patients while sparing GCS-associated side effects in children [13]. Excluding offending dietary agents, immune and/or microbial modulation, and improved intestinal barrier by decreasing inflammation are thought to explain the mechanisms of EEN action [14]. Detecting the predictors of response to EEN can help elucidate mechanisms, identify diet therapy candidates, and improve current treatments, which is especially important given challenges with EEN therapy such as taste fatigue, tolerability, and significant costs [15].

EEN-induced microbial shifts differ between therapy responders and non-responders [16] and may impact therapy response. Previously, long-term diet was identified in animal studies to predict microbial shifts in response to dietary therapy [17]. Given the stability of the microbiome despite changes in response to short-term dietary alterations and the ability for baseline microbiota to predict response to EEN [16], the microbiome established in response to a patient’s long-term diet may play a role in mediating the response to subsequent EEN therapy [18]. This is supported by models where the gut microbiome is identified as more important than the food itself for predicting clinical outcomes to foods such as glycemic response [19]. As diet is a modifiable factor, showing a predictive role can positively and safely impact treatment outcomes for patients with IBD. We therefore hypothesized that in pediatric patients with CD, pre-diagnosis long-term dietary patterns may predict responses to EEN, mediated through the intestinal microbiome.

## 2. Materials and Methods

### 2.1. Setting and Participants

The Canadian Children Inflammatory Bowel Disease Network (CIDsCaNN; https://cidscann.ca/ (accessed on 28 March 2024)) prospectively enrolled and followed new-onset pediatric IBD cases until transition to adult services, as previously described [20]. Comprehensive baseline phenotypic and longitudinal disease activity assessments, laboratory parameters, and therapy details were collected prospectively at ad hoc and routine six-month periodic reviews using standardized case report forms (CRFs) and documented on REDCap [21]. IBD-focused pediatric gastroenterologists approved the diagnostic IBD label using conventional clinical, endoscopic, and histologic criteria according to the Paris classification [22], with disease location based on macroscopic findings observed via colonoscopy and MR enterography. Endoscopic findings were documented using the Simple Endoscopic Score for CD (SES-CD).

### 2.2. Patient and Public Engagement

Patients/caregivers of pediatric IBD patients and other stakeholders have identified their most important research questions; their third priority research question was, “What role does diet have in the management of pediatric IBD?” [23]. These priorities have informed our research directions and hypothesis.

### 2.3. Study Design

This was an open-label prospective study within CIDsCANN. Eligible children were newly diagnosed with CD and enrolled between February 2014 and June 2017, treated with EEN as a first induction therapy, and completed food frequency questionnaires (FFQs, reflecting the year prior to diagnosis) within 90 days of diagnosis. From the national cohort of ~1500 children newly diagnosed with IBD, 942 had CD (following the Porto criteria) [24]; 103 of these patients met the eligibility criteria by having an FFQ available and receiving EEN as their primary induction therapy. Parents completed FFQs for younger children as needed. Patients from 10 pediatric centers across Canada met the inclusion criteria (Appendix A). Informed consent/assent was obtained from parents/patients, and participants were free to withdraw at any time. Ethics approval was obtained at each participating center (University of Alberta Ethics ID#: Pro00042980).

Fecal calprotectin (FCP) was measured by locally validated ELISA at baseline, 2, 12, and 18 months (Appendix A). Clinical outcomes were collected every 6 months for 18 months following EEN initiation.

The FFQ was previously validated in Canadian pediatric and adolescent populations and assessed dietary intakes over the previous 12 months [25]. Dietary pattern analysis was conducted using principal component (PC) analysis (PCA) on kilocalorie-adjusted food groups (Appendix A); the obtained PCs were used to identify dietary patterns that maximally captured dietary variability within our study population. These data-derived diet patterns capture correlated food consumptions within our study population and are named for the food themes that most characterize each pattern (see further details in the Appendix A). Patient scores for each PC reflected their relative adherence to each dietary pattern (for each patient, an adherence value for each of the four patterns was obtained). Importantly, each patient included in the study had a calculated rate of adherence (defined relative to the entire study population) to each of the dietary patterns identified, and each individual was highly or poorly adherent to more than one diet. To assess associations with an MED, we used a tertile-defined scoring system to calculate a relative MED (rMED) score for each patient. Established by Buckland et al. [26] based on Trichopoulou et al. [27], 1–3 points were assigned reflecting relative tertiles of intake after calorie adjustment by the Willett residuals method (referenced and summarized in the Appendix A). To calculate rMED scores, three points were assigned to patients in the highest tertiles of vegetables, fruits, nuts, legumes, fish, cereals, and red wine, as well as those in the lowest tertiles for consumption of high-fat dairy and meat. Two and one points were assigned to the declining tertiles, respectively.

### 2.4. Clinical Outcomes of Interest

Clinical features, including weighted pediatric CD activity index (wPCDAI) and physician global assessment (PGA) [28], were recorded at baseline and every 6 months. Failure of EEN was defined as the need for therapy with either monoclonal anti-tumor necrosis factor (TNF)-α antibodies (mostly infliximab) or systemic GCS therapy prior to completing 60 days of EEN. Therapy decisions were not protocolized but rather determined by the treating physician. Reasons for stopping EEN were individually reviewed (by SD and EW), based on systematic data entry; failures were sub-classified into “lack of response” (replaced EEN therapy within 14 days) or “loss of response” (replaced EEN after 14–60 days). Survival time until disease flare or need for treatment escalation were defined by a need for GCS/anti-TNF, respectively, after successful EEN. A reason for stopping cited as “patient choice” was classified as intolerance of EEN (considered neither response nor failure). Administration of immunomodulators (methotrexate; azathioprine) or anti-TNF as pre-planned maintenance therapy at any time point was recorded but not defined as failure. SES-CD [29] was obtained at diagnosis and used as a baseline indicator of disease activity. Disease location, defined by the Paris classification, was categorized as ileal CD (Paris L1) or colonic/ileocolonic CD (Paris L2/L3); due to small numbers, L4 was not specified [22]. Improvement in PGA was defined as any reduction in PGA by six months.

### 2.5. Fecal Microbiome 16s rRNA Sequencing Analysis 

Stool specimens were collected and frozen immediately in the patient’s freezer, transported on ice, and stored at −80 ± 10 °C. Bacterial DNA was extracted from an aliquot (two to three punch biopsies from frozen stool; 0.18–0.22 g) of fecal samples as described previously and sequenced with minor modifications [30]. Bioinformatics was completed using the QIIME2 [31] pipeline with DADA2 [32], generating amplicon sequence variants (ASV) used for further statistical analysis. DNA extraction, sequencing, and bioinformatics are further described in the Appendix A.

### 2.6. Statistical Analysis

Dietary patterns resulting from dimensionality reduction of dietary data by PCA and rMED scores were correlated with clinical features such as age, sex, disease location, post-treatment six-month wPCDAI [33], and PGA scores. Mann–Whitney U tests, Kruskal–Wallis tests, and Spearman correlations were completed using Stata 14 (Stata Statistical Software: Release 14. StataCorp LP, College Station, TX, USA) [34]. Multivariable COX proportional hazard regression survival analysis of time to anti-TNF or disease flare was performed using Stata 14 [34], with patients censored at their last recorded appointment if <18 months of follow-up. Sex, disease location, and perianal disease showed no significant interactions (*p* > 0.1) and so were not included in the model. When assessing six-month height Z-scores, patients taking GCS prior to their six-month follow-up were excluded. Simpson and Shannon diversity and Chao1 richness were calculated using ASV-level microbial data. Data were visualized using GraphPad Prism version 9.1.2 for Windows (GraphPad Software, San Diego, CA, USA; www.graphpad.com).

## 3. Results

### 3.1. Patient Clinical Characteristics 

One hundred and three patients met the inclusion criteria (EEN induction therapy and completed FFQ within a mean of 24 days) with complete baseline data (Figure 1A); eligible patients were diagnosed February 2014–March 2017. Five patients concomitantly beginning anti-TNF induction therapy were excluded, leaving 98 patients with a median of 518 days of follow-up [interquartile range (IQR) 109]. Twelve (12%) patients failed EEN induction therapy (required GCS or unplanned anti-TNF therapy within 60 days), four of whom had a lack of response (4%, failure <14 days), while eight had a loss of response (8%, failure 14–60 days). Seventeen patients (17%) experienced disease flare (required GCS) during the 18 months of follow-up, four of whom also met the criteria for EEN failure (included in both groups). Two patients (2%) were intolerant of EEN. Seventy-six (76%) patients successfully completed EEN induction therapy and did not experience a disease flare during follow-up; however, 48/76 (63%) EEN responders were initiated on anti-TNF maintenance therapy during follow-up. Overall, 40 out of 103 (39%) patients were on anti-TNF maintenance therapy by six months, and 79 (77%) were on immunomodulators.

The mean age at enrollment and diagnosis was 12.5 years (standard deviation: 2.8 y), with no significant difference between males and females (*p* > 0.5; Figure 1B). No difference in disease location or complicated disease were seen between males and females, and 96% had uncomplicated inflammatory disease (Paris B1). Forty-eight patients had a six-month PGA available and had not required GCS or anti-TNF (used to assess long-term disease activity in response to EEN therapy), showing significant improvement (Figure 1C). wPCDAI was only available for 23 patients at six months, with 22 baseline and six-month pairs. wPCDAI in this small subset was not associated with clinical outcomes of interest or microbiota. Baseline SES-CD score was available for 89 patients.

A total of 36 patients (37%) provided stool for 16S analysis at baseline (prior to starting therapy) and 24 at six months, with 18 pairs. Sixteen and fourteen stool specimens for 16S were collected at 12 and 18 months, respectively (Appendix A). FCP was available at baseline for 29 patients and after two months for 13 patients, with nine pairs. There was a trending decrease in FCP in those eight patients with successful EEN induction (*p* = 0.055, *n* = 8; Figure 1D).

### 3.2. Dietary Pattern Associations with Patient Features

#### Dietary Pattern Analysis

Adjusted servings were used to conduct PCA that was orthogonally rotated to minimize correlation between the different dietary factors. Patient scores for each PC were converted into Z-scores for correlation with clinical outcomes. Z-scores greater than 3 or less than −3 were deemed outliers and were excluded from further analysis. A cut off of four PCs/dietary patterns was used based on eigenvalues (Appendix A). Dietary factor loadings > 0.2 were considered of importance in the first four PCs and used to characterize and name the dietary patterns (Table 1). Of note, each individual included in the study will have a value of adherence to each of the four identified dietary patterns.

Four dietary patterns (PAs) had eigenvalues > 3, and each accounted for >7% of dietary variability (Appendix A). Food groups with high factor loadings that characterized each pattern are summarized in Table 1. Food groups with positive correlations (factor loadings) had increased consumption with increased adherence to the pattern, while food groups with negative correlations had decreased consumption with increased adherence to the pattern. The “vegetarian” pattern was characterized by high consumption of whole grains, vegetable soup, soy and tofu products, salad dressing, fruit, full-fat dairy, and butter, with relatively low intakes of fried or skin-on chicken or turkey. The “meat” food pattern included more rice products, non-vegetable soups, and both red and non-red meats, with relatively low intakes of unfried or skin-off chicken or turkey and granola bars. “Pre-packaged” had high intakes of high-fiber cereals, sugary condiments, breaded fish, and diet soda, with relatively low intakes of processed and lean red meat. Finally, the “mature” pattern had relatively high intakes of unfried or skin-off chicken or turkey, fish, seafood, vegetables, fruit, coffee, alcohol, and milk alternatives, with a relatively low intake of pizza.

rMED scores were positively associated with vegetarian and mature diet adherence (rho = 0.2035, *p* < 0.05; rho = 0.5989, *p* < 0.0001, respectively) and had a trending positive association with adherence to a pre-packaged pattern (rho = 0.1907, *p* = 0.0537; Appendix A). Age was positively associated with mature diet adherence (rho = 0.3586, *p* < 0.001; Appendix A). Mature diet adherence and rMED score were negatively associated with SES-CD score in females (*n* = 37, rho = −0.42, *p* < 0.05; rho = −0.4406, *p* < 0.01, respectively); mature diet adherence was also negatively associated with baseline FCP in males (rho = −0.4556, *p* < 0.05, *n* = 20) (Figure 2A–C).

### 3.3. Dietary Pattern Associations with Clinical Outcomes

Males exhibiting a loss of clinical response to EEN had significantly higher meat diet adherence (*n* = 4, median 0.82, vs. responders *n* = 51, median −0.17, IQR −0.57–0.38; Figure 2E, *p* < 0.05). Males with higher meat diet adherence also trended to fail EEN induction therapy (lack/loss of response, *p* = 0.075). Meat diet adherence was not associated with disease flare (need for GCS) in male patients or with any clinical outcomes in female patients. Failure numbers were insufficient to control for baseline disease activity. Among the subset of 48 patients with PGA scores available at both baseline and 6 months, rMED score was associated with a decreased likelihood of PGA improvement and a smaller reduction in PGA from baseline to six months (rho = −0.4406, *p* < 0.01; *p* < 0.05, respectively; Figure 2D,F) but was not associated with our defined EEN clinical outcomes (e.g., flare or need for anti-TNF). Only 5 of 48 patients (10%) showed no improvement in PGA over 6 months. rMED score also trended lower in males (*p* = 0.068), who also tended (*p* = 0.056) to have less severe disease at baseline. Numbers of patients that had PGA available at both baseline and 6 months were insufficient to perform sub-analysis by sex.

Surprisingly, a pre-packaged diet was associated with protection or delay of disease flare in male patients. Males in the lowest tertile for pre-packaged diet adherence experienced earlier disease flares [*p* < 0.05, (relative risk) RR = 0.392, 95% CI 0.186–0.827; Figure 2G], but this might reflect the association of the pre-packaged dietary pattern with rMED (Appendix A); pre-packaged diet adherence did not predict EEN failure or a need for anti-TNF in males and was not associated with clinical outcomes in female patients. For patients with successful EEN induction, a higher mature diet adherence was associated with a delayed need for anti-TNF (*p* = 0.086, RR = 0.634; 95% CI 0.377–1.067, Figure 2H). Neither vegetarian diet adherence nor rMED scores were associated with our defined EEN treatment outcomes, disease flare timing, or need for anti-TNF. To illustrate the differences in dietary intake between tertiles of adherence for each dietary pattern, Appendix A compare differences in daily servings of food groups.

### 3.4. Baseline and 6-Month Microbial Composition and Diversity Are Associated with Pre-Diagnosis Diet

After identifying dietary links with treatment outcomes, we assessed for associations between fecal microbiota and diet. Patterns of microbiota, as identified by 16SrRNA sequencing and summarized by PCA, were evaluated for correlation with dietary pattern adherence and rMED scores. Mature diet adherence and rMED scores were found to correlate with patient baseline intestinal microbiota patterns (rho = −0.4266, *p* < 0.05; rho = −0.4415, *p* < 0.01, respectively). 

The microbiota amplicon sequences variants (ASVs) most strongly represented in the correlation with mature diet and rMED scores at baseline (largest factor loadings) were three Faecalibacterium ASVs, two Blautia, Ruminococcus gnavus group ASVs, Ruminococcus torques, Ruminococcus gauvreauii groups, Coprococcus, and Ruminococcaceae UCG-002. Among these, one Blautia and one Faecalibacterium ASV were associated with higher mature diet adherence (rho = 0.3989, *p* < 0.05; rho = 0.4801, *p* < 0.01, respectively), while lower mature diet adherence was associated with an increased Ruminococcus gnavus group ASV (rho = −0.4179, *p* < 0.05, Figure 3A–C). The second Blautia and Faecalibacterium ASVs were not significantly associated with mature diet adherence (Figure 3A–C). Higher rMED scores were associated with increased Blautia (ASV1: rho = 0.4004, *p* < 0.05; ASV2: rho = 0.4635, *p* < 0.01, Figure 3D). Two Faecalibacterium ASVs were approaching significance (ASV1) and significantly (ASV2) associated with higher rMED scores (ASV1: rho = 0.3273, *p* = 0.0514; ASV2: rho = 0.4322, *p* < 0.01, Figure 3E). The Ruminococcus gnavus group ASV was negatively associated with rMED scores (rho = −0.4444, *p* < 0.01, Figure 3F). Ruminococcus torques, Coprococcus, and Ruminococcaceae UCG-002 were not associated with any of the dietary patterns or rMED scores.

Higher mature diet adherence was associated with increased Shannon (but not Simpson) diversity and richness (Chao1) at baseline (rho = 0.3989, *p* < 0.05; rho = 0.3952, *p* < 0.05; rho = 0.4968, *p* < 0.01, respectively; *n* = 35 with available microbiome data, Figure 3G–I). Higher vegetarian diet adherence was associated with increased microbial Simpson diversity and increased Shannon diversity (rho = 0.3583, *p* < 0.05; rho = 0.3323, *p* = 0.0508, respectively; *n* = 35), but was not significantly associated with richness (Chao1, rho = 1347, *p* > 0.05, *n* = 35, Figure 3J–L). rMED scores were positively associated with Simpson and Shannon diversity as well as microbial richness (Chao1) at baseline (rho = 0.3302, *p* < 0.05; rho = 0.4454, *p* < 0.01, *n* = 36; rho = 0.5218, *p* < 0.01, respectively; *n* = 36, Figure 3M–O).

Higher mature diet adherence was also associated with increased Simpson diversity, trending Shannon diversity, and richness (Chao1) at six months (rho = 0.4675, *p* < 0.05; rho = 0.4234, *p* = 0.0559; rho = 0.3891, *p* = 0.0813, respectively, Figure 4A–C). Pre-packaged diet adherence was inversely associated with Shannon diversity (but not Simpson) and richness at six months (rho = 0.3984, *p* < 0.05; rho = 0.5059, *p* < 0.01, respectively, Figure 4D–F). Meat diet adherence was not associated with any significant differences in microbial diversity.

### 3.5. Lower Baseline Microbial Diversity Was Associated with Earlier Disease Flare

There were no significant differences in baseline microbiota patterns or diversity/richness scores in patients that failed EEN compared to responders (loss or lack of response; failures *n* = 4, responders *n* = 27; the five patients who started anti-TNF therapy concomitant with EEN were not included). Among those who responded to EEN (*n* = 27 with baseline stool available), lower Simpson and Shannon microbial diversity at baseline was associated with earlier disease flare (HR:0.00044, *p* < 0.05, 95%CI 0–0.819; HR = 0.403, *p* = 0.05, 95%CI 0.162–1.002, Figure 5). Richness (Chao1) was not associated with disease flare. For graphical purposes, tertiles of microbial diversity are shown in Figure 5; continuous diversity was used for analyses.

### 3.6. Additional Sex-Specific Findings

We identified several sex-specific correlations between diet, microbes, and clinical outcomes. Adherence to diet patterns (Appendix A) was significantly different: males showed a higher adherence to the mature diet (median −0.13, IQR 1.38; females median −0.15, IQR 1.06; *p* < 0.05) with higher rMED scores (males median 17, IQR 14.5–19; females median 15, IQR 13–18; *p* = 0.068). Females had a significantly higher PGA (disease severity) at baseline (*p* < 0.05, Appendix A). There was no significant difference in PGA at six months between males and females (Appendix A). Baseline SES-CD score and FCP showed sex-specific correlations with dietary patterns (Figure 2E), and as described above—increased adherence to the meat dietary pattern was associated with lack of clinical response to EEN only in males (*p* < 0.05, Figure 2E). Additionally, only among males was higher pre-packaged diet pattern adherence associated with delayed disease flare (*p* < 0.05, Figure 2G). In females, significant differences in diet were associated with disease location: mature diet adherence was higher for Paris L1 (median 0.23, IQR 1.42; Paris L2/L3 0.036, IQR 1.22, *p* < 0.01), and pre-packaged diet adherence was lower for Paris L1 (median −0.11, IQR 0.89; Paris L2/L3 −0.037, IQR 0.57, *p* < 0.01, Appendix A).

## 4. Discussion

Prospective data and stool samples collected from children newly diagnosed with CD who received EEN as primary induction therapy allowed us to investigate if pre-diagnosis dietary patterns could predict EEN treatment response and outcomes over 18 months of follow-up. In addition, we assessed for variations in intestinal microbes with diet and treatment response. We found that pre-diagnosis diet was indeed associated with some important clinical outcomes, including response to EEN, disease flare, and a need for therapy escalation to anti-TNF. As summarized in Figure 6, we also identified diet–microbial diversity and microbial diversity–treatment/outcome associations, suggesting that gut microbiota (as established by diet) may indeed mediate responses to EEN.

Our identified dietary patterns were named vegetarian, meat, pre-packaged, and mature (derived from detailed patient FFQ data) and are consistent with other Canadian pediatric population-derived dietary patterns [35], as well as dietary patterns identified by meta-analysis in other populations [36].

Our finding of an inverse correlation between baseline disease severity, mature adherence, and rMED scores is supportive of other findings suggesting a protective effect of the MED in IBD [9]. This is further supported by our identified association between increased mature diet adherence and a delayed need for anti-TNF. Although not associated with our defined EEN outcomes of interest, rMED was associated with a decreased likelihood of PGA improvement and a smaller reduction in PGA from baseline to six months. Analysis of this significance is complicated, as males had lower rMED scores and lower baseline disease severity. Additionally, two-thirds of our population was male, but there were insufficient numbers of patients with both baseline and 6-month PGAs available to perform sub-analysis by sex, as very few patients (5/48, 10%) did not see an improvement in PGA. It remains possible that there is a relationship between a MED and less improvement of long-term outcomes with EEN, apart from a need for anti-TNF or GCS, while a MED remains protective in IBD prior to EEN administration. Given that less than half of our patients (48/103) had data available for long-term PGA analysis, this relationship requires further investigation.

Our observed association between increased meat diet adherence and EEN induction failure supports reports linking increased meat consumption to more active disease [8], possibly through meat-related intestinal microbiome alterations, as shown in IBD animal models [37,38]. Higher microbial diversity associated with our identified vegetarian diet (low in meat consumption) also suggests a role for microbial mediation, as decreased α-diversity is a hallmark of IBD [6].

Surprisingly, a higher pre-packaged diet adherence was associated with delayed disease flare in males. This could reflect protection from the decreased consumption of lean red and processed meat found with this diet or the consumption of high-fiber cereals (increased in this dietary pattern), resulting in increased protective fiber intake [39]. Pre-packaged adherence also trended towards a positive association with rMED scores, suggesting that this dietary pattern captures multiple complex relationships, including a more Mediterranean style of eating. PCA to obtain data-derived dietary patterns captures actual consumption patterns, which can result in clustering of seemingly counteracting food products. Although this complicates the identification of single culprit or savior foods, it accurately captures the complexity of patients’ diets, which contain a diversity of so-called healthful and ultra-processed foods. Holistic analysis of patients’ diets in observational studies, as we have performed here, can provide clues for future experimental studies to identify causal relationships. Taking this into account, it is important to remember that our findings cannot directly support or disprove a causative role for diet in mediating IBD pathogenesis.

Decreased microbial diversity has been previously associated with increased disease severity in pediatric IBD [40]. Our findings that a lower baseline microbial diversity was associated with earlier disease flare (Figure 5) illustrate that microbial changes could precede a clinically detectable need for GCSs. These findings support the role of microbial mediators in the pathogenesis of IBD and can therefore provide further targets in the development of novel microbe-altering therapies. 

Mature diet adherence was associated with increased microbial diversity and richness at baseline and six months, as well as a trending association with delayed need for anti-TNF (Figure 2H) among those with successful EEN induction. Increased microbial diversity has been found to predict treatment outcomes with anti-TNF in pediatric IBD [40]. Furthermore, suggesting a microbial mediator for these dietary associations, mature diet adherence was associated with a decrease in *Ruminococcus gnavus* group ASVs, which are strict anaerobic gram-positive bacteria that have been associated with increased disease severity in pediatric IBD [41]. Mature diet adherence and rMED scores were positively associated with baseline *Blautia* and *Faecalibacterium* ASVs. Some strains of *Faecalibacterium* are thought to be beneficial due to their ability to produce butyrate, which has been associated with lower intestinal inflammation and greater intestinal barrier integrity [42,43]. Increased dietary intake of fish, nuts, fruits, vegetables, and cereals (similar to a MED) are associated with higher abundance of *F. prausnitzii* [43]. Increasing fruit and vegetable intake is recommended in CD to reduce bacterial mucin metabolism and increase short-chain fatty acid (SCFA) production through microbial fiber fermentation [39]. Increased fruit and vegetable intake with higher adherence to the mature diet could result in increased dietary fiber, providing a substrate for beneficial commensal microbiota.

*Blautia* present a more complicated picture—although increased baseline *Blautia* spp. can predict a lower likelihood of remission on EEN [16], Hart et al. found increased *Blautia* in pediatric IBD patients that achieved remission on EEN or GCSs [44]. Additionally, increased *Blautia* has been found in patients that achieved remission with the Crohn disease exclusion diet (CDED) dietary therapy [45]. Characterization to the species and strain levels, along with functional assessment, is likely necessary to clarify their complex relationship with host health. 

Sex-specific associations between diet and risk for IBD have been previously identified [7], as well as sex-specific alterations in the microbiome in response to dietary changes [46]. The average age of our cohort was 12.6 years; post/intra-pubertal levels of sex hormones may help explain these associations. As our cohort was predominantly male (59.2%), it is possible that we had increased power to detect significant relationships in males, although some findings were observed in females (e.g., association between “pre-packaged” and “mature” dietary patterns and disease location). These results may also be partly explained by gender/cultural factors, but need to be interpreted with caution due to multiple comparisons and low power. One potential consequence of note: in some cohorts, pediatric patients with ileal CD (L1) have been found more likely to succeed with EEN than those with colonic disease (L2/L3) [47]. Perhaps most significantly, females had more severe disease at baseline, and it is possible that dietary influences on the microbiome are overshadowed by the greater effects of inflammation on the microbiome.

Lastly, although diet after diagnosis and completion of EEN was not evaluated, our identified associations between pre-diagnosis diet and 6-, 12-, and 18-month microbiota (Figure 4) supports either long-term stability of the established microbiome despite a temporary change in diet for EEN therapy (as supported by others) [18] or a correlation between pre-diagnosis and post-EEN diet; this highlights the potential ability for long-term diet to continuously influence host–microbe interactions in lifelong diseases such as IBD.

### Limitations

While our study produced multiple interesting associations, these need to be interpreted with caution. Although our FFQ did not provide the granularity to assess micronutrients such as fiber, whole foods are more closely correlated with the microbiome than individual nutrients [5], supporting our investigation of associations between dietary patterns and treatment outcomes with potential mediation by the microbiome. FFQs are prone to over-reporting and biases such as social desirability; our use of kilocalorie adjustment and relative dietary patterns within our population help minimize (but do not eliminate) these effects. While most of the FFQs were collected close to diagnosis, some were collected several weeks later; patients were asked to fill the FFQs to best reflect their pre-diagnosis diet, but it is possible that pre-diagnosis diet was already impacted by subclinical symptoms. It is important to note that we do not have data on dietary habits after EEN therapy, and we cannot assume that the diet is the same or different; changes in dietary behavior after nutritional therapy have been observed by others [48]. 16S rRNA gene sequencing is limited to genus-level phenotypic characterizations of bacteria and does not assess other components of the intestinal microbiome (e.g., microbial function); therefore, our findings provide only initial insight into relationships with microbiota abundance. Fungi and other gut microbes were not assessed.

Smaller numbers of stool samples sizes reduced our power to identify relationships with microbiota and FCP (especially given the low number of EEN failures) and limited our ability to adjust for baseline disease activity. Short-term treatment response at EEN completion was not routinely collected, so we indirectly assessed treatment failure through the need for additional medications and longer-term outcomes (i.e., six-month visit), similar to outcome assessments in other studies [49]. Furthermore, objective measures of remission (endoscopy or FCP) were not available for most patients, so response to EEN was measured clinically and not by follow-up endoscopic assessment. Longer-term outcomes (e.g., 6 months) were also affected by additional therapies and are not necessarily a direct result of EEN. Although there are consensus guidelines for CD treatment and clinicians in our study specified the reason for each therapy, some subjectivity remains between clinicians and sites in a multi-center study. The very low rate of EEN failures limited the power of our study to identify predictors of negative outcomes. Finally, multiple outcomes and relationships between many variables were assessed, in some cases with small numbers, leading to numerous results with variable statistical significance; although this can increase the risk of type I error, this exploratory study of complex associations provides us with direction for new hypotheses for further investigation. Therefore, many of these results need to be further confirmed in larger studies to best interpret our findings.

## 5. Conclusions

Multi-center prospective sampling and data collection enabled us to identify dietary predictors of treatment response and long-term outcomes in pediatric patients with CD receiving EEN induction therapy. A mediating role of the microbiome is supported by diet–microbiota and microbiota–clinical outcome associations. For example, a “mature diet” can decrease disease severity by increasing beneficial microbes (such as *Faecalibacterium*). Our findings suggest that baseline microbiota established through long-term diet partly determines therapy response; this may be considered in evaluation for future treatment regimens and development of personalized diets. As shown in Figure 6, further investigations of the complex relationships identified here can allow for more targeted therapies that will better harness diet-microbe-disease interactions. 

## Figures and Tables

**Figure 1 nutrients-16-01033-f001:**
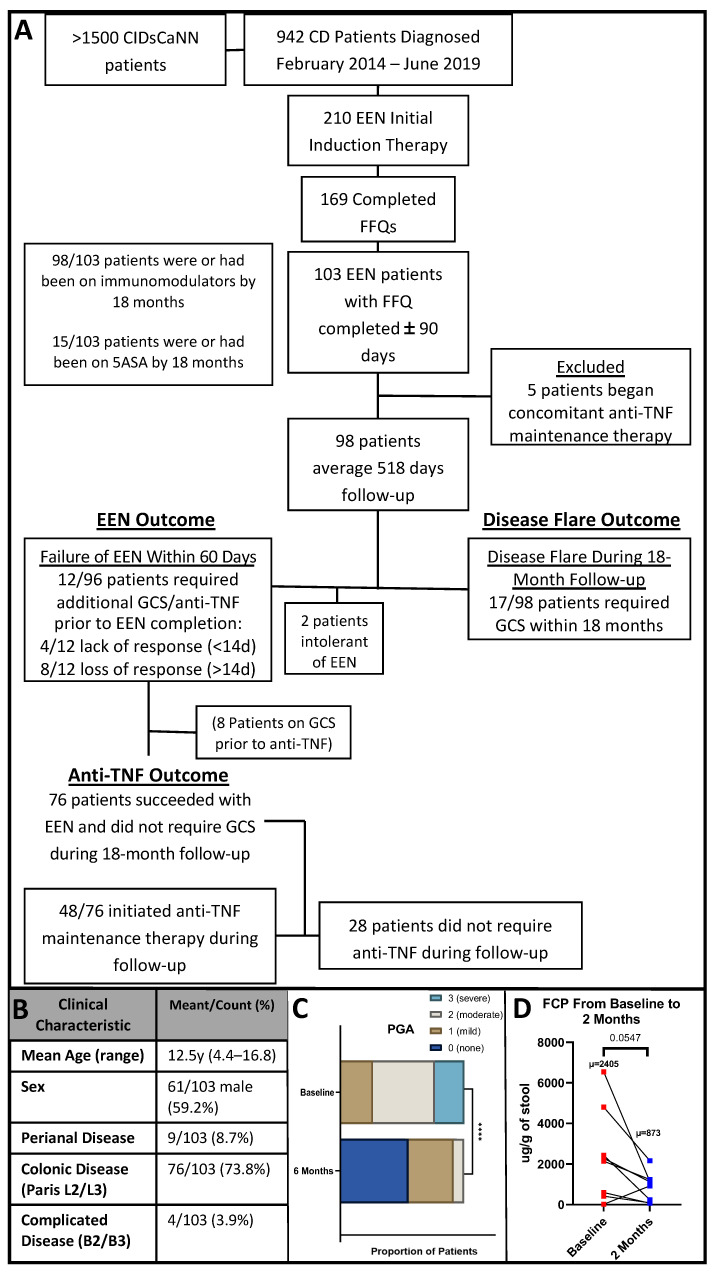
Patient characteristics and major outcomes. (**A**): Flowchart of patient numbers throughout inclusion and outcome criteria. After exclusion, 98 patients were included in the analysis, of whom 76 did not require steroids, but 48 of those were initiated on anti-TNF therapy during maintenance. EEN = exclusive enteral nutrition, 5ASA = 5-aminosalicylate, anti-TNF = anti-tumor necrosis factor-α, GCS = glucocorticoid. (**B**): Baseline clinical characteristics of 103 patients included in the study. (**C**): Physician global assessment (PGA) was available for 48 patients at baseline and 6 months; PGA improved from baseline to 6 months follow-up. (**D**): Fecal calprotectin (FCP) tended to decrease from baseline to 2 months for the eight patient responders with paired specimens. **** = *p* < 0.0001.

**Figure 2 nutrients-16-01033-f002:**
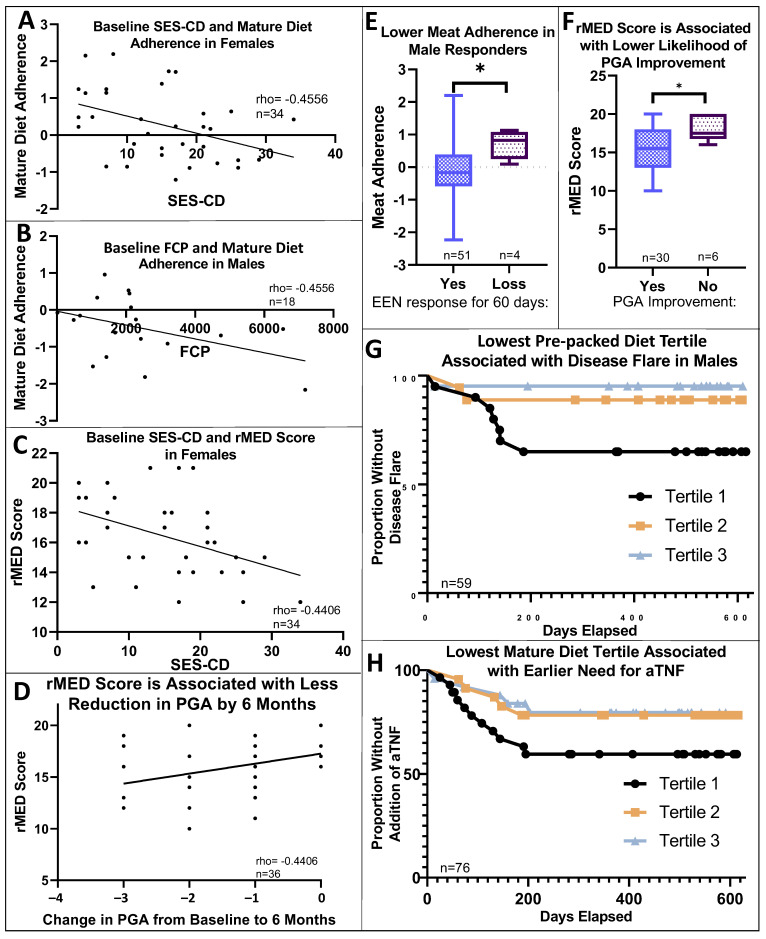
Dietary associations with baseline and long-term clinical outcomes. (**A**): Mature diet adherence was negatively associated with SES-CD score at baseline in females. (rho = −0.4556, *p* < 0.05, n = 34 with relevant data available). (**B**): Mature diet adherence was negatively associated with baseline FCP in males (rho = −0.4556, *p* < 0.05, *n* = 18). (**C**,**D**): rMED score was negatively associated with SES-CD score at baseline and showed a smaller reduction in PGA in females. (rho = −0.42, *p* < 0.05, *n* = 34, rho = −0.4406, *p* < 0.01, *n* = 36, respectively). (**E**): Higher meat adherence was associated with a loss of response to EEN in males (*p* < 0.05). (**F**): A higher rMED score was associated with a decreased likelihood to show improvement in PGA from baseline to six months. (**G**): Lower adherence to a pre-packaged diet was associated with earlier development of disease flare in males, *p* < 0.05, *n* = 59. (**H**): Lower adherence to a mature diet was associated with an earlier need for anti-TNF, *p* = 0.086, *n* = 76. * = *p* < 0.05, tertile 1 = lowest adherence, tertile 3 = highest.

**Figure 3 nutrients-16-01033-f003:**
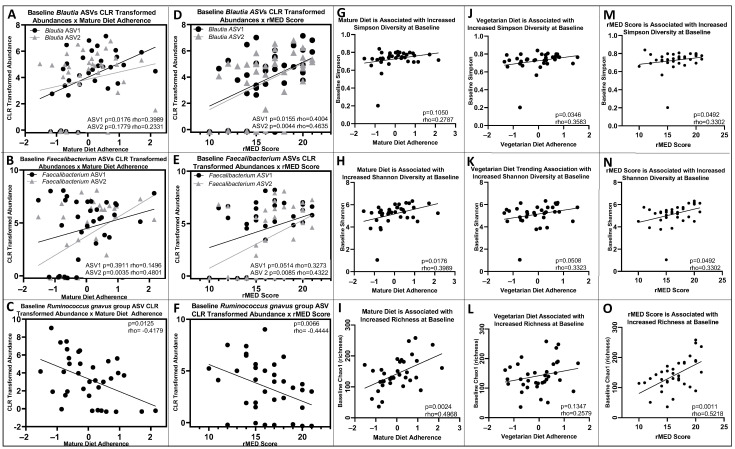
Dietary patterns and rMED score are associated with baseline microbial abundances, microbial diversity, and richness. (**A**,**B**): Baseline Blautia ASV1 and Faecalibacterium ASV2 relative abundances were positively associated with mature diet adherence (rho = 0.3989, *p* < 0.05, *n* = 35; rho = 0.4801, *p* < 0.01, *n* = 35). Blautia ASV2 and Faecalibacterium ASV1 were not significantly associated (*p* > 0.05). (**C**): Baseline Ruminococcus gnavus group ASV relative abundance was negatively associated with mature diet adherence (rho = −0.4179, *p* < 0.05, *n* = 35). (**D**): Baseline Blautia relative abundances were positively associated with rMED score (ASV1 rho = 0.4004, *p* < 0.05, *n* = 36; ASV2 rho = 0.4635, *p* < 0.01, *n* = 36). (**E**): Faecalibacterium ASV1 baseline relative abundance was positively associated with rMED score at baseline (rho = 0.3273, *p* = 0.0514, *n* = 36), ASV2 baseline relative abundance was positively associated with rMED score (rho = 0.4322, *p* < 0.01, *n* = 36). (**F**): Ruminococcus gnavus group ASV baseline relative abundance was negatively associated with rMED score (rho = −0.4444, *p* < 0.01, *n* = 36). (**G**–**I**): Simpson diversity at baseline was not associated with mature diet adherence, but mature diet adherence was associated with increased Shannon diversity and richness (Chao1) at baseline (rho = 0.2787, *p* > 0.05, *n* = 35; rho = 0.3989, *p* < 0.05, *n* = 35; rho = 0.4968, *p* < 0.01, n = 35). (**J**–**L**): Increased vegetarian diet adherence was associated with increased Simpson and trending Shannon microbial diversity, but not richness (Chao1), at baseline (rho = 0.3583, *p* < 0.05, *n* = 35; rho = 0.3323, *p* = 0.0508, *n* = 35; rho = 0.1347, *p* > 0.05, *n* = 35). (**M**–**O**): rMED scores were positively associated with Simpson and Shannon diversity as well as microbial richness (Chao1) at baseline (rho = 0.3302, *p* < 0.05, *n* = 36; rho = 0.4454, *p* < 0.01, *n* = 36; rho = 0.5218, *p* < 0.01, *n* = 36).

**Figure 4 nutrients-16-01033-f004:**
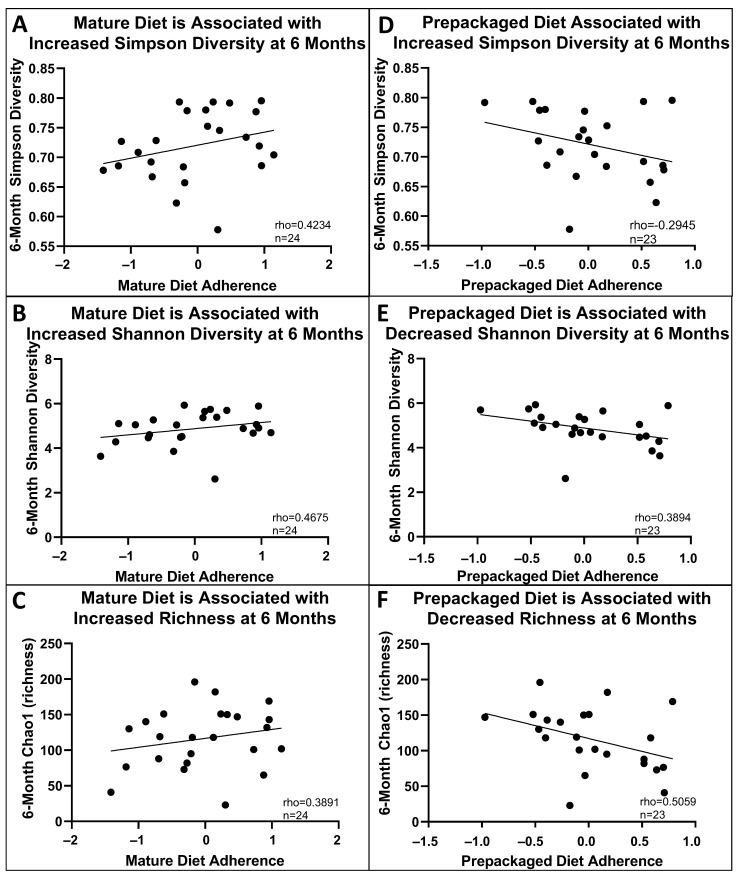
Pre-diagnosis diet is associated with microbial diversity at 6 months. (**A**–**C**): Increased mature diet adherence was associated with increased Simpson diversity, increased Shannon diversity, and an increase in richness at 6 months (rho = 0.4234, *p* = 0.0559, *n* = 24; rho = 0.4675, *p* < 0.05, n = 24; rho = 0.3891, *p* = 0.0813, *n* = 24). (**D**–**F**): Pre-packaged diet adherence was not associated with Simpson diversity at 6 months (rho = −0.2945, *p* > 0.05) but was inversely associated with Shannon diversity and richness at 6 months (rho = 0.3984, *p* < 0.05, *n* = 23; rho = 0.5059, *p* < 0.01, *n* = 23).

**Figure 5 nutrients-16-01033-f005:**
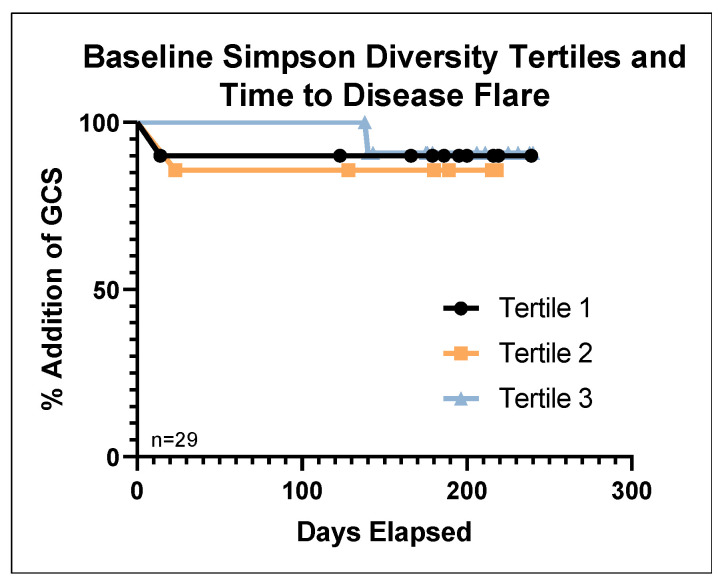
Lower baseline microbial diversity is associated with decreased time to disease flare. A lower Simpson diversity at baseline was associated with decreased time until disease flare requiring GCS; tertiles are shown for illustrative purposes; *n* = 29, with three disease flares. Tertile 1 = lowest diversity, tertile 3 = highest.

**Figure 6 nutrients-16-01033-f006:**
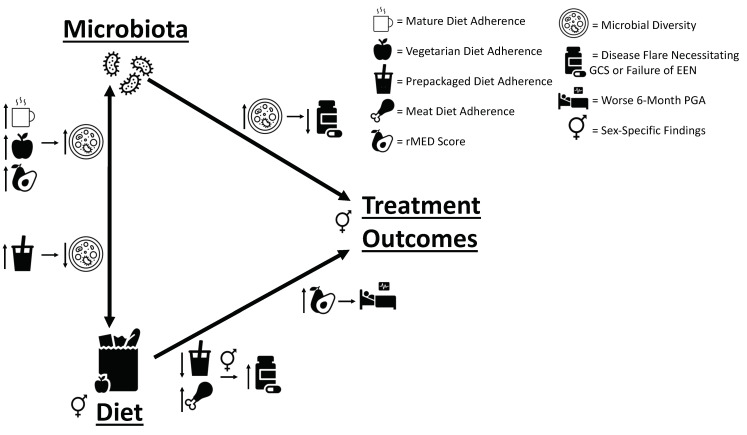
Linking pre-diagnosis diet to treatment outcomes through fecal microbiota. Our study revealed multiple links between pre-diagnosis diet, microbes, and patient outcomes, which are summarized in this figure. Mature and vegetarian diets and rMED score were positively associated with baseline microbial diversity, while pre-packaged diet was negatively associated with six-month diversity. Increased baseline diversity was associated with prolonged survival until disease flare. A decreased pre-packaged diet adherence and increased meat diet adherence were associated with decreased time until disease flare and lack of EEN response, respectively. Finally, increased rMED score was associated with a worse six-month PGA and a decreased likelihood for PGA improvement from baseline.

**Table 1 nutrients-16-01033-t001:** Main features of identified dietary patterns.

Dietary Pattern:	“Vegetarian”	“Meat”	“Pre-Packaged”	“Mature”
**Food Groups Positively Associated (factor loading)**	Whole Grains 0.3048	Rice, Rice Noodles, Couscous 0.2887	High-Fiber Cereals 0.3944	Chicken, Turkey without skin/fried 0.207
	Vegetable Soup 0.3134	Other Soups 0.4105	Sugary Condiments 0.2856	Fish 0.2581
	Soy/Tofu 0.3047	Red Meat 0.277	Breaded Fish 0.3549	Seafood 0.2148
	Salad Dressing 0.2733	Pork 0.3495	Diet Soda 0.4256	Vegetables 0.3751
	Fruit 0.2103	Liver Organs 0.2043		Fruit 0.2546
	Full-Fat Dairy 0.2678	Chicken, Turkey with skin or fried 0.3161		Coffee 0.2245
	Butter 0.2973			Alcohol 0.2503
				Milk Alternatives 0.2153
**Food Groups Negatively Associated (factor loading)**	Chicken, Turkey with skin or fried −0.222	Chicken, Turkey without skin/fried −0.2104	Lean Red Meat −0.2424	Pizza −0.3615
		Granola Bars −0.2191	Processed Meat −0.2182	

Dietary patterns obtained as the first four principal components from FFQ PCA analysis. Food groups and their factor loadings (representing the relative contribution of each food group to the dietary pattern) are shown. Green shading represents food groups with a positive association with dietary adherence scores; red shading indicates food groups with a negative association with dietary adherence scores.

## Data Availability

The 16S raw sequencing data are deposited into the Sequence Read Archive (SRA) of NCBI (http://www.ncbi.nlm.nih.gov/sra) under BioProject PRJNAXXXX. Other data underlying this article cannot be shared publicly for the privacy of individuals that participated in the study.

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
