# Peer review of "Pre-Diagnosis Diet Predicts Response to Exclusive Enteral Nutrition and Correlates with Microbiome in Pediatric Crohn Disease"

_nutrients, 2024, doi:10.3390/nu16071033_

Round 1
Reviewer 1 Report
Comments and Suggestions for Authors
Summary: The paper by Dijk S et al presents a prospective multi center cohort of children with newly diagnosed Crohn’s disease to be treated with 6-8 weeks of exclusive enteral diet. The purpose is to evaluate whether diet for 1 year before diagnosis impacts on response to EEN (failure, relapse and need for biologic therapy e.g. TNFα). Clinical endoscopic and intestinal microbiome characteristics were evaluated. These occurred at baseline, and six months (as well as at 12 and 18 months). Clinical features were assessed by the pediatric CDAI, and Physician global assessment score and SES-CD (at baseline). Fecal microbiome was evaluated using genus level 16SrRNA. Biodiversity is evaluated by calculating the Shannon (biased richness) and Simpson (biased evenness) Index and richness is evaluated also by using the Chao1 index. Diet is assessed by a food frequency questionnaire and weighing scores for adherence to groups of diets based on Principal Component Analysis.
The results support an association between pre diagnosis diet clinical outcome, to some extent SES-CD an dmicrobiome changes between baseline as well as at 6 months. In addition certain sex related alterations are found. From the initial 103 children 98 with variations of available data are included. Four food patterns are found, Vegetarian and a Mature diet are associated with an rMED score (anti- inflammatory). On the other hand a meat diet is not associated with rMED while the Prepackaged diet is variable. Baseline microbiome was associated with type of diet and this was also true to some extent at 6 months.
Two suprising findings were made. One that a high rMED score at baseline was associated with a worse PGA and a smaller improvement at 6 mo. The second unexpected finding was that a prepackaged diet in men delayed flares perhaps it was still closer to a mediterranean type diet (Fig S1).
General comments: I think this is an interesting study and does reveal a connection between pre diagnosis diet and response to EEN. However, I found the result section quite difficult to follow because there is a large number of analytical outcomes, many trending only. Also there were few failures and relapses in the follow up period. As a result there are a lot of statistical comparisons. The discussion is easier to follow.
Indeed this is a pilot which needs to be confirmed and reproduced. In particular the sex dependent outcome as pointed out by the authors could be a function of bias.
Pg 8/20,Table 1 Is it possible to assign numbers to each diet group? I found the use of n values very confusing. On pg 10/20 line 285 Lower adherence to “ prepackaged “ meal was associated with earlier disease flare. So ti is unclear whether of these 59 males also adhered to a Mature diet?
This may also be relevant because on pg 11/20 lines 341 -349 it is stated that there were 35 patients who adhered to a vegetarian diet and another 35 adhered to a Mature diet?
In addition the hypothesis to account of the benefit of the pre-packaged diet for men may have been a resemblance of the diet to a Mediterranean type Fig S1 (low inflammatory). It seems there is an overlap in diet intake in groups but this analysis is not clear.
The lack of clarity could raise a hypothetical question if not properly interpreted. If the above is correct i.e. it means that over 71% of children followed a relatively low inflammatory diet and yet they still got Crohn’s disease suggesting that the diet is more of a modulator of disease (rather than a direct cause). It also raises a question about early dietary exposure since the patients were children?
A minor criticism is the way the rho and p values are reported. It may be better to state the rho value first then the p statistic
Pg 9/20, line 267 please check the p value ? should be 0.05 not 0.5
Pg 10/20, lines 283, 284. The thinking in this paper is that the pre-diagnostic diet is resumed after 6-8 weeks of EEN and followed out to 18 months? If this is the case why would a high rMED predict less improvement after initial EEN? Since a high rMED is associated with beneficial bacteria and predicted a better response to EEN.
Author Response
We would like to thank the reviewer for the very helpful comments, which we are convinced help us improve this manuscript and clarify some of the complex aspects of our paper.
Below is a point-to-point response to the comments:
Reviewer 1:
Summary: The paper by Dijk S et al presents a prospective multi center cohort of children with newly diagnosed Crohn’s disease to be treated with 6-8 weeks of exclusive enteral diet. The purpose is to evaluate whether diet for 1 year before diagnosis impacts on response to EEN (failure, relapse and need for biologic therapy e.g. TNFα). Clinical endoscopic and intestinal microbiome characteristics were evaluated. These occurred at baseline, and six months (as well as at 12 and 18 months). Clinical features were assessed by the pediatric CDAI, and Physician global assessment score and SES-CD (at baseline). Fecal microbiome was evaluated using genus level 16SrRNA. Biodiversity is evaluated by calculating the Shannon (biased richness) and Simpson (biased evenness) Index and richness is evaluated also by using the Chao1 index. Diet is assessed by a food frequency questionnaire and weighing scores for adherence to groups of diets based on Principal Component Analysis.
The results support an association between pre diagnosis diet clinical outcome, to some extent SES-CD and microbiome changes between baseline as well as at 6 months. In addition certain sex related alterations are found. From the initial 103 children 98 with variations of available data are included. Four food patterns are found, Vegetarian and a Mature diet are associated with an rMED score (anti- inflammatory). On the other hand a meat diet is not associated with rMED while the Prepackaged diet is variable. Baseline microbiome was associated with type of diet and this was also true to some extent at 6 months.
Two suprising findings were made. One that a high rMED score at baseline was associated with a worse PGA and a smaller improvement at 6 mo. The second unexpected finding was that a prepackaged diet in men delayed flares perhaps it was still closer to a mediterranean type diet (Fig S1).
General comments: I think this is an interesting study and does reveal a connection between pre diagnosis diet and response to EEN. However, I found the result section quite difficult to follow because there is a large number of analytical outcomes, many trending only. Also there were few failures and relapses in the follow up period. As a result there are a lot of statistical comparisons. The discussion is easier to follow.
Response: We thank the reviewer for the excellent summary of our findings and for highlighting some of the challenges with the way the results were presented. We have addressed some of these issues below and have further emphasized these in the limitations section (lines 670-674): ‘The very low rate of EEN failures limited the power of our study to identify predictors of negative outcomes. Finally, multiple outcomes and relationships between many variables were assessed, in some cases with small numbers, leading to numerous results with variable statistical significance;…’.
Indeed this is a pilot which needs to be confirmed and reproduced. In particular the sex dependent outcome as pointed out by the authors could be a function of bias.
Response: We completely agree with this comment and have added the following statement to the end of the limitations section (lines 676-677): ‘Therefore, many of these results need to be further confirmed in larger studies to best interpret our findings.’
Pg 8/20, Table 1 Is it possible to assign numbers to each diet group? I found the use of n values very confusing. On pg 10/20 line 285 Lower adherence to “prepackaged “ meal was associated with earlier disease flare. So ti is unclear whether of these 59 males also adhered to a Mature diet?
Response: The reviewer raises an important point – it seems that we have not completely clarified how some of our data are presented. Each individual will have variable adherence rates to each of the 4 diets, so adherence to one diet does not mean that you are not adherent (or less adherent) to another. The definition of adherence to each diet is relative to the entire group of 98 individuals included in the study, so one patient can be on the 79%ile adherence to Mature diet, 53%ile to vegetarian, 91% to Meat, and 32% to Pre-packaged. Therefore, adding an N to Table 1 would not be informative since all patients contributed to this analysis and all will have a rate of adherence to each diet. The ‘n’ value in Figure 2 represents the number of individuals included in the analysis (for example, n=59 means that we had data on flare for 59/61 males included in the study). To further clarify this point, we have added the following to the Methods section (lines 137-139): ‘Importantly, each patient included in the study will have a calculated rate of adherence (defined relative to the entire study population) to each of the dietary patterns identified, and each individual is highly or poorly adherent to more than one diet.’ and this to the Results section (lines 246-247): ‘Of note, each individual included in the study will have a value of adherence to each of the 4 identified dietary patterns’. Finally, we have added to the legend to Figure 3 ‘with relevant data available’ after the first mention of ‘n’ (line 292).
This may also be relevant because on pg 11/20 lines 341 -349 it is stated that there were 35 patients who adhered to a vegetarian diet and another 35 adhered to a Mature diet?
Response: We have again clarified that the n=35 means that microbiome data were available for 35 individuals (line 365).
In addition the hypothesis to account of the benefit of the pre-packaged diet for men may have been a resemblance of the diet to a Mediterranean type Fig S1 (low inflammatory). It seems there is an overlap in diet intake in groups but this analysis is not clear.
Response: We thank the reviewer for raising this point. Indeed, and as we hope better explained now, there can be and is overlap between the dietary patterns, and Fig S1 does indeed show that the Mediterranean diet overlaps with more than one dietary pattern. The specific point raised that the higher Pre-packaged diet adherence was associated with delayed disease flare in males could be partially explained by the association with rMED scores is mentioned in the discussion already (lines 580-582: ‘Pre-packaged adherence also trended a positive association with rMED scores, suggesting that this dietary pattern captures multiple complex relationships, including a more Mediterranean style of eating.’) but we have also added the following clarification to the Results section (line 321-322 ‘…but this might reflect the association of the Pre-packaged dietary pattern with rMED (Figure S1)’.
The lack of clarity could raise a hypothetical question if not properly interpreted. If the above is correct i.e. it means that over 71% of children followed a relatively low inflammatory diet and yet they still got Crohn’s disease suggesting that the diet is more of a modulator of disease (rather than a direct cause). It also raises a question about early dietary exposure since the patients were children?
Response: To address this comment, we have added the following sentence to the discussion (lines 589-591): ‘Taking this into account, it is important to remember that our findings cannot directly support or disprove a causative role for diet in mediating IBD pathogenesis’.
A minor criticism is the way the rho and p values are reported. It may be better to state the rho value first then the p statistic
Response: We thank the reviewer for this comment and have reversed the order of rho and p value throughout the paper.
Pg 9/20, line 267 please check the p value ? should be 0.05 not 0.5
Response: Thank you for noticing this error; it is now corrected.
Pg 10/20, lines 283, 284. The thinking in this paper is that the pre-diagnostic diet is resumed after 6-8 weeks of EEN and followed out to 18 months? If this is the case why would a high rMED predict less improvement after initial EEN? Since a high rMED is associated with beneficial bacteria and predicted a better response to EEN.
Response: The reviewer raises a very important point. We do not know (or claim) that patient diet resumes to pre-diagnosis diet after completing EEN; in fact, some data from other studies suggests that IBD patients might change their dietary habits after receiving dietary therapy (e.g., Nutrients. 2023 Jan 20;15(3):554. doi: 10.3390/nu15030554.PMID: 3677126). Unfortunately, we do not have any data on actual dietary habit of the patients included in this study after EEN therapy. Therefore, one might speculate that a ‘less healthy’ diet prior to diagnosis (low rMED score) might be associated with significant opportunity for improvement and better outcomes (and therefore, high rMED might be associated with less PGA improvement) but this is just speculation.
To clarify that we do now have data on post-EEN diet, we have added the following sentence to the Discussion (lines 654-656): ‘It is important to note that we do not have data on dietary habits after EEN therapy and we cannot assume that the diet is the same or different; changes in dietary behaviour after nutritional therapy have been observed by others {Martín-Masot, 2023 #7813}’.
Reviewer 2 Report
Comments and Suggestions for Authors
I would like to congratulate the authors for their manuscript. I just finished reading a very interesting article regarding the association of pre-diagnosis diet with EEN outcomes in children with CD.
The study is designed very well with appropriate and coprehensive presentation of the results. The discussion is in accordance with the results.
I have only to comments to make.
In line 456 of the Discussion section you mention that your results can be generalized globally. Do you mean the results of the cited meta-analysis or yours? Yours cannot be generalized globally, and as far as a meta-analysis is concerned, a meta-analysis may build statistical power and may increase generalizability by including studies conducted in multiple settings with different samples.
The second comment I would like to make is that no objective method such as colonoscopy for remission took place, but as long as you mention that in the limitations of your study, I don' t need any response or change in your manuscript regarding this.
Author Response
We would like to thank the reviewer for the very helpful comments, which we are convinced help us improve this manuscript and clarify some of the complex aspects of our paper.
Below is a point-to-point response to the comments:
Reviewer 2:
Comments and Suggestions for Authors
I would like to congratulate the authors for their manuscript. I just finished reading a very interesting article regarding the association of pre-diagnosis diet with EEN outcomes in children with CD.
The study is designed very well with appropriate and comprehensive presentation of the results. The discussion is in accordance with the results.
Response: We thank the reviewer for their very positive comments!
I have only to comments to make.
In line 456 of the Discussion section you mention that your results can be generalized globally. Do you mean the results of the cited meta-analysis or yours? Yours cannot be generalized globally, and as far as a meta-analysis is concerned, a meta-analysis may build statistical power and may increase generalizability by including studies conducted in multiple settings with different samples.
Response: Thank you for this important comment, and we agree that our statement was not clear. We meant to say that the dietary patterns we identified show similarity to those identified in other studies (and not that our results can be generalized). We have changed this sentence to now read as follows: ‘…as well as dietary patterns identified by meta-analysis in other populations’ (lines 552-553).
The second comment I would like to make is that no objective method such as colonoscopy for remission took place, but as long as you mention that in the limitations of your study, I don' t need any response or change in your manuscript regarding this.
Response: We completely agree with this comment; indeed, we did mention in the Limitations section of the Discussion that ‘objective measures of remission (endoscopy or FCP) were not available for most patients so response to EEN was measured clinically’ but have now added the following to the end of this sentence (lines 668): ‘…and not by follow up endoscopic assessment’.